# Physical Stability of Lotus Seed and Lily Bulb Beverage: The Effects of Homogenisation on Particle Size Distribution, Microstructure, Rheological Behaviour, and Sensory Properties

**DOI:** 10.3390/foods13050769

**Published:** 2024-03-01

**Authors:** Jiajia Su, Xiaokun Qiu, Yi Pei, Zhuo Zhang, Guanghui Liu, Jiaojiao Luan, Jiangli Nie, Xihong Li

**Affiliations:** 1College of Horticulture and Landscape Architecture, Tianjin Agricultural University, Tianjin 300384, China; jjiasu@126.com (J.S.); qxk333333@163.com (X.Q.); peiyee@126.com (Y.P.); zhuozhang1999@163.com (Z.Z.); 18526134867@163.com (G.L.); l18353650242@163.com (J.L.); 2State Key Laboratory of Food Nutrition and Safety, College of Food Science and Engineering, Tianjin University of Science and Technology, Tianjin 300457, China

**Keywords:** homogenisation, particle size distribution, microstructure, rheological behaviour, sedimentation index, lotus seed, lily bulbs

## Abstract

The lotus seed and lily bulb beverage (LLB) has a problem with solid particle sedimentation. To address this issue, LLB was homogenised twice at different pressures (0~100 MPa) using a homogeniser. This study aims to investigate the changes in the particle size distribution (PSD), microstructure, rheological behaviour, sedimentation index (IS), turbidity, physicochemical properties, and sensory quality of LLBs after homogenisation treatments. The results regarding PSD and microstructure showed that the suspended particles were decomposed at high pressure with increasing homogenisation pressure, forming small particles of cellular material, cell wall fragments, fibre fractions, and polymers. The LLB showed shear-thinning behaviour and weak gelation characteristics (G′ > G″) and rheological properties. Among all homogenisation pressures, the 60 MPa sample showed the lowest sedimentation rate and the highest turbidity. When the pressure was increased from 0 to 100 MPa, the total soluble solid (TSS) content showed an upward trend, while the ascorbic acid content (AAC) gradually decreased. The highest sensory evaluation was observed in the 60 MPa sample in terms of overall acceptability.

## 1. Introduction

Nowadays, with the increasing concern for health management and disease prevention, there is growing consumer demand for nutritious and tasty natural plant-based functional beverages, which are less lactose intolerant than dairy products. These beverages are made from a variety of ingredients, including fruits, vegetables, grains, nuts, and seeds [1,2]. Therefore, the development of nutrient-rich functional plant-based beverages has a promising future in the nutraceutical market, potentially impacting public health and nutritional status [3]. Lily (*Lilium brownii* var. *viridulum*) is a herbaceous bulb plant of the Liliaceae family, native to China, and its medicinal history can be traced back to Shen NonG′s herbal classic of the Han Dynasty [4]. It is rich in proteins, vitamins, and a variety of bioactive components, such as total phenolic content (2000 mg·100 mL^−1^), total flavonoid content (150 mg·100 mL^−1^), total flavanol content (66 mg·100 mL^−1^), etc. Its edible organ is mainly the bulb, which has the health benefits of relieving cough, lowering blood sugar, antitumour properties, and improving immunity [5]. Lotus seeds and lotus roots are the main edible parts of the lotus (*Nelumbo nucifera* Gaertn.). The seeds also have a history of about 3000 years in China, and its components include carbohydrates (61–62%), total protein (16–21%), and trace elements such as calcium (30–31 mg·100 mL^−1^), sodium (30–35 mg·100 mL^−1^), and iron (13–18 mg·100 mL^−1^). Lotus seeds are widely used in daily diet and medicine, with antioxidant, anti-inflammatory, anticancer, and diabetes prevention effects [6].

The combination of lotus seeds and lily bulbs is a traditional Chinese medicinal diet; it is usually used to cook porridge or make desserts with high medical and edible values. The two ingredients are rich in flavonoids and alkaloids, which are effective in treating mental anxiety, insomnia, and irritability [7]. However, fresh lily bulbs have a long growth period but a short harvesting period, which makes them prone to mechanical damage due to their high water content and vulnerable surface. In long-term storage, the epidermis is prone to browning, rotting, mildew, and other problems, resulting in the limited development of the lily bulb industry [8]. The combination of lotus seeds and lily bulbs to make plant-based functional beverages can not only make it easier for consumers to obtain a nutritious traditional medicinal diet suitable for all ages but also reduce the problem of lily storage difficulties, which is a very potential development strategy. However, lotus seeds and lily bulbs are rich in starch, which leads to the tendency for solid particles to settle when they are made into beverages, and such settled particles affect the level of sensory acceptance by consumers. Therefore, the sedimentation index of the beverage is an essential parameter that is influenced by particle size, particle distribution, and total soluble solids. Homogenisation is used to solve the problems of the flocculation and precipitation of suspended particles [9].

Homogenisation technology could represent a promising strategy for changing the physical properties of the liquid and significantly improving the particles’ stability. The term homogenisation means that the liquid is forced to undergo pressure through a homogeniser so that the particles suspended in the liquid appear in a uniform distribution [10]. Homogenisation treatment distorts particles, fluid cells, and molecules into nanoscale particles by creating high shear stresses through extreme pressure and narrow space. The fragmentation of particles interacts with shear actions, turbulence, and cavitation during homogenisation, leading to the dispersion of individual elements. Moreover, homogenisation has a variety of functions, such as particle size reduction, mixing, dispersion, and emulsification [11,12]. Salehi et al. showed that homogenisation could reduce the particle size of juices (tomato, taro, cashew, apple, blueberry, etc.) to sub-micrometres through pressure, thus changing the sedimentation, PSD, and rheological properties of the pulp [13]. Niccolò et al.’s study on chia seeds showed that homogenisation treatment reduced the particle size and changed protein conformation [14]. Wellala et al. found that homogenisation improved the rheological properties and cloud stability and reduced microbial populations in blended juices consisting of apple, peach, and carrot [15]. Luo et al. found that the treatment of quinoa protein isolates with 50 MPa homogenisation resulted in the destruction of large protein aggregates, a reduced particle size, enhanced gel properties, and a more homogeneous microstructure [16]. Homogenisation treatment can improve the sensory qualities of liquid foods without affecting their nutritional values. Suárez-Jacobo et al. used homogenisation to treat apple juice and found that the antioxidant capacity of apple juice was not changed [17]. In addition, homogenisation can effectively extend the shelf life and reduce the microbial activity of fruit juice [18]. Donsì et al. mentioned that when juices were treated at 100 MPa, the homogenised juice improved in flavour and texture, while the untreated juice deteriorated within 20 d [19]. Homogenisation is also used to emulsify unstable liquids such as soy-protein-stabilised emulsions [20], condensed milk, ice cream, and other dairy products to improve their flavour and texture while extending their shelf life [21]. He et al. showed that homogenisation was very effective on plant milk made from oats, adzuki beans, and adlay [22].

However, in terms of plant-based functional beverage processing, no studies have been published on the effects of homogenisation for beverages made from lotus seeds and lily bulbs. This study aimed to determine the effects of different homogenisation treatments on the physical properties and sensory qualities of a LLB, including rheological properties, particle size, microstructure, colour change, and nutritional properties. The result will provide a reference for selecting homogenisation conditions for the LLB and provide theoretical support for its development.

## 2. Materials and Methods

### 2.1. Raw Materials

Dried lotus seeds (5 kg) were purchased from Jin Yuanbao Market (Tianjin, China). Lily bulbs (5 kg) were harvested in Qilihe District, Lanzhou City, Gansu Province, a famous lily growing area with more than 400 years of cultivation history, which produces Lanzhou lily, the only sweet lily for both medicinal and food uses in China. Uniform-sized fresh lily bulbs were dug on 15 August 2023 and transported to the laboratory via a cold chain. The raw materials were kept in cold storage (4 °C) for subsequent use.

### 2.2. Product Development

Dried lotus seeds without cores were selected, which were washed and soaked in tap water for 2 h before being pre-cooked at 95 °C for 15 min until the fruit softened. Fresh and high-quality lily bulbs were selected and washed, with the mud removed, the bulbs broken, and heated at 95 °C for 5 min. The pretreated materials (25 g of lotus seeds and 25 g of lily bulbs) were mixed with food additives (7.5% white granulated sugar, 0.2% ascorbic acid, 0.18% citric acid, 0.06% CMC, and 0.08% Xanthan gum), and 450 mL of water was added. Then, the mixture was ground using a high-speed blender (PB12 × 1, Midea Living Appliance Manufacturing Co., Ltd. Wuhu, China) for 1 min and filtered twice with double-layer gauze. All samples were stored in a 500 mL sterile high-temperature resistant polypropylene plastic bottle in a refrigerator (4 °C) for future use.

### 2.3. Homogenisation Treatment of LLB

Before homogenisation treatment, the homogeniser (BP-3/100, Dao You Industrial Equipment Co., Ltd., Shanghai, China) was sterilised with 70% ethanol and washed twice with hot water. The samples were removed from the refrigerator and immediately loaded into the homogeniser, where they were homogenised twice under different pressures at 0 (control), 20, 40, 60, 80, and 100 MPa. After homogenisation, the sample temperature was measured using a thermometer, and the results showed that the outlet temperature of the samples increased with homogenisation pressure (i.e., 0 MPa, 20.50 ± 0.50 °C; 20 MPa, 28.50 ± 1.32 °C; 40 MPa, 37.30 ± 0.58 °C; 60 MPa, 51.00 ± 0.87 °C; 80 MPa, 64.80 ± 0.29 °C; and 100 MPa, 73.80 ± 1.04 °C). All the homogenised samples were put back into the bottle and pasteurised at 70 °C for 20 min to ensure that they were safe to eat, which were then cooled down at room temperature and stored in a refrigerator at 4 °C after sterilisation.

### 2.4. Particle Size Distribution

The particle size parameters were measured using a laser particle size analyser (Bettersize 3000, Baxter Instrument Co., Ltd. Dandong, China). The particle size was distinguished by the volume-based diameter (D [4,3]) and area-based diameter (D [3,2]) [23].

### 2.5. Microstructure Observation

The micrographs of LLB particles were carried out using a SU1510 scanning electron microscope (Sulin Scientific Co., Ltd. Hubei, China) according to the method proposed by Sharma et al. [23]. Briefly, 3 mL of the samples were freeze-dried for 48 h (−50 °C) in an FD-IA-50 lyophilizer (Bilang Co., Ltd. Shanghai, China). After grinding into powder, they were glued to a sample table with double-sided tape, and their PSDs were observed using an electron microscope.

### 2.6. Rheological Properties

A rheological test was carried out using a rotational rheometer (MARS 60, Haake Technik GmbH, Vreden, Germany) following the methodology described by Saricaoglu et al. with some modifications [24]. The rheometer was equipped based on parallel plate geometry with the shape (diameter: 35 mm), whose gap size was set at 2 mm. The test was conducted at a constant temperature (25 °C), before which 70% ethanol was utilised for disinfection, and then a small amount of the samples were transferred using a pipette and moved to the centre of the rheometer plate to prevent the formation of air bubbles while minimising the damage to the sample structure.

Equation (1) expresses the relationship between the shear rate and shear stress and the shear rate and apparent viscosity in steady-state rheological tests:(1)$$\tau =K\times \gamma^{n}$$
where *τ* is the shear stress (Pa), *K* is the consistency coefficient (Pa·sn), *γ* is the shear rate (s^−1^), and *n* is the flow behaviour index.

For dynamic rheology, we used frequency sweep to determine the viscoelastic behaviour of the LLB samples at a temperature of 25 °C and a scanning frequency (1~100 Hz) to determine the viscoelastic characteristics of the test samples by comparing G′ (elasticity) and G″ (viscosity) at different frequencies.

### 2.7. The Sedimentation Test and Sedimentation Index

The sedimentation index was determined according to the methodology described by Staubmann et al. [25]. The samples were kept in 10 mL sealed glass bottles, placed upright in a 4 °C refrigerator, and remained stable and undisturbed except when transported from the refrigerator to the experiment table. The heights of the bottom sediment, middle turbid liquid, and top supernatant in sealed glass bottles were measured with a vernier calliper at 0, 48, and 96 h after the sample preparation. An analysis of the sedimentation index was carried out as previously reported [24] with some modifications. The 10 mL of samples were placed in centrifuge tubes and subjected to centrifugation at 4000 rpm for 20 min using a high-speed freezing centrifuge (TGL-16M, Scenery Technology Co., Ltd. Ji’an, China). The supernatant was carefully decanted after centrifugation, and the mass of the sediment was measured. The sediment degree of the samples was evaluated according to the index of sedimentation, which was calculated using the following Equation (2):(2)$$Sedimentation\;Index=\frac{M_{0}}{M}\times 100$$
where *M*_0_ is the sedimentation volume (g); *M* is the total sample volume (g).

### 2.8. Turbidity Measurement

The turbidity (%) was determined according to the methodology described by Zhong et al. [26]. The LLB samples were centrifuged (8000 rpm, 10 min), and the supernatant was extracted into a 96-well microplate. The absorbance values with wavelengths ranging from 650 nm to 700 nm were measured using a microplate reader (SpectraMax190, Precision Instrument Co., Ltd. Yunze, China).

### 2.9. Colour Parameters

The colour parameters were determined using a colourimeter (SR-66, Ledi Instrument Co., Ltd. Ningbo, China) according to the methodology described by Kubo et al. [23]. The LLB samples were shaken evenly and poured into a glass cuvette to measure their L*, a*, and b* values after being corrected with black and white plates. The colour variation (∆E*) was calculated using the following Equation (3):(3)$$\Delta E=\sqrt{{\left(L\;-\;L_{0}\right)}^{2}+{\left(a\;-\;a_{0}\right)}^{2}+{\left(b\;-\;b_{0}\right)}^{2}}$$
where the *L*_0_, *a*_0_, and *b*_0_ values refer to the colour of the samples under 0 MPa, while L, a, and b are the colour of the samples under 20, 40, 60, 80, and 100 MPa.

### 2.10. pH and Total Soluble Solids (TSS)

The pH was measured using a pH instrument (PHS-3E, Yiqian Scientific Instruments Co., Ltd. Xinghua, China). After calibration with a buffer solution, a pH meter probe was inserted into the sample liquids to determine the pH value. A PAL-α digital refractometer (AK002B, Ceyou Technology Co., Ltd. Shenzhen, China) was used to measure the TSS (◦Brix).

### 2.11. Ascorbic Acid Content (AAC)

The ascorbic acid (mg·100 mL^−^^1^) was determined using the 2,6-dichlorophenol indophenol titration method described by Santana et al. [27]. The samples (10 mL) were placed into a 100 mL volumetric flask and titrated to scale with a 20 g·L^−^^1^ oxalic acid solution, which was shaken and extracted for 10 min. The extraction solution (10 mL) was absorbed into a 100 mL triangular bottle and titrated until it appeared reddish without fading for 15 s.

### 2.12. Sensory Evaluation

The sensory evaluation was determined using Pali et al.’s method [28]. The appearance, colour, aroma, mouthfeel, taste, and overall acceptability of the LLB were scored by a sensory evaluation group involving 10 participants. The sensory evaluation should be in an area without smell or noise, the temperature should be controlled within 20~22 °C, the relative humidity should be maintained at 50~55%, and the room should be draughty. Appropriate amounts of samples were poured into 50 mL open transparent containers and placed in a refrigerated environment at 4~6 °C, which should not be co-stored with others that are toxic, harmful, smelly, or have adverse effects on them. The samples were taken out only before the evaluation began so that their temperature during the evaluation was within the range of 6~10 °C. The result was based on an intensity scale from −4 to 4 points, where the differences were defined as very substantial (−4 and 4), considerable (−3 and 3), significant (−2 and 2), minimal (−1 and 1), or not significant (0 points) compared to the samples under 0 MPa. Algebraic symbols, i.e., negative or positive, indicate lower or higher perceptions [29].

### 2.13. Statistical Analysis

All experiments were arranged using a randomised design and conducted in triplicates. Data were reported as means ± standard deviation (SD) for triplicate determinations. The ANOVA analysis of variance was used to compare the mean values, the SPSS 20.0 statistical analysis was adopted, and all graphs in this study were plotted using Origin 2018. The level of significance was set at (*p* < 0.05).

## 3. Results

### 3.1. Particle Size Distribution (PSD)

Figure 1 illustrates the effect of several homogenisation pressures on the particle size distribution of LLB samples. As shown in Figure 1a, the average particle size of LLBs is reduced after the homogenisation treatment, which is consistent with the observations in previous studies on fruit and vegetable products, such as lily pulps [30], rosehip nectar [24], tomato juice [31] and blackcurrant juice [32]. Gul et al. found that cavitation, friction, shear, and turbulence phenomena were generated after homogenisation, and as the homogenisation pressure increased, suspended particles such as cellular debris, polymers, and fibrous particles in the samples were further broken down into smaller sizes [33].

In addition, as shown in Table 1, the samples treated under 0~100 MPa showed the phenomenon of unimodal distribution. Compared with the sample under 0 MPa, the PSD span of those under 20 and 40 MPa were significantly reduced, which represented the width of the PSD, with smaller values indicating a narrower distribution [23]; however, the samples under 60~100 MPa had a significantly higher span than those under 0~40 MPa (*p* < 0.05). This is contrary to the findings of Liu et al. [30], who reported a narrower distribution of PSD with increasing pressure on homogenised lily pulps from 20 to 100 MPa, possibly because two composite raw materials, lily bulbs and lotus seeds, were used in our study and that homogenisation preferentially crushed one of the more friable materials, resulting in a slightly larger measured span value. Moreover, the particle diameter decreased while the number of particles increased after homogenisation treatment, which increased the number of contact points among the particles due to mechanical and chemical interactions [23]. On the other hand, the span parameter was determined by the cumulative distribution values D10, D50, and D90 (Table 1). Therefore, the proportion of these three values greatly influenced the parameter of span. In this study, the average particle size of LLBs decreased with the increase in pressure, and the cumulative distribution values of the particles of samples treated under 60~80 MPa decreased accordingly so that the derived parameter of the span was the one that represented the wider distribution of LLBs in the range of small particles.

The effect of the homogenisation treatment on the LLB volume (D [3,2]) and surface weight (D [3,2]) is shown in Figure 1b. The values of both D [4,3] and D [3,2] significantly decreased (*p* < 0.05) due to the homogenisation pressure, with a reduction of 78% and 82% in D [4,3] and D [3,2] in the range of 0~60 MPa, while a reduction of 23% and 15% in D [4,3] and D [3,2] occurred in the range of 60~100 Mpa, respectively. This result indicated that both large and small particles were greatly affected when the samples were treated under less than 60 MPa, while large and small particles were less affected when the pressure was higher than 60 MPa, but large particles were more seriously damaged than small particles. The damage behaviour of particles under homogeneous pressures appeared asymptotic. Yu et al. also reported similar patterns of homogenisation effect on particles [34]. As explained by Augusto et al. about the effect of homogenisation on suspended particles, the remaining cells were destroyed during the homogenisation process, and the fragments in the liquids were broken up into smaller particles [31]. However, these small fragments were less susceptible to fragmentation than slightly larger or whole cells.

### 3.2. Microstructure

Figure 2 describes the micrographs of LLBs at different high pressures, and it is obvious that there are many large spherical particles of irregular shapes in the sample under 0 MPa, with small particles and inhomogeneous suspensions distributed around them. The sizes of the particles in the samples under 20~40 MPa were gradually unified but still showed scattered small particles dispersed in the suspension. When the pressure increased to 60~100 MPa, the particles therein became more rounded, with fewer and fewer broken particles visible.

In order to further analyse the microstructure of LLBs, we used a scanning electron microscope to observe it. As shown in Figure 3, the structural morphology of the sample under 0 MPa changes drastically compared to those under 20~100 MPa. The surface of the samples under 0 MPa is rough, aggregated, and stacked, showing irregular geometry shapes, with scattered small particles visible around them; these accumulated aggregates may be formed by some complete proteins in cells or plants after lyophilization and grinding; meanwhile, irregular geometric structures are formed, while smaller particles are related to starch particles, fibre particles, or other substances [35]. In addition, samples under 20~100 MPa stretch after homogenisation treatment, gradually forming irregular lamellar structures with sharp edges and smooth surfaces, which are fragmented into smaller particles. As observed through a particle size analysis, these suspended particles were strongly decomposed under high pressure, leading to the formation of small particles such as cellular materials, cell wall fragments, fibre fractions, and polymers. Similar microstructural changes were obtained by Huang et al. in their study on homogenisation-treated sugar beet pulps (5~100 MPa) [36]. In fact, homogenisation has been evaluated to affect microstructural changes not only in many fruit and vegetable juices or products such as pumpkin [37] and mango juices [38] but also in a number of protein beverages, such as peanut milk [39] and skim milk powder [40]. These results indicated that homogenisation strongly affected the particle size, particle number, and microstructural changes in suspensions.

### 3.3. Rheological Characterisation

#### 3.3.1. Steady-State Shear Properties

Rheological properties reflect the deformation and flow behaviour of substances. The rheological behaviour is mainly related to the types of interactions between the particles and molecules responsible for gel formation [41]. The effect of different homogenisation pressures on LLBs was described through the response of the shear rate and shear stress as well as the shear rate and apparent viscosity in rotation mode (Figure 4). The shear stress of all samples increased with the shear rate, while the apparent viscosity decreased with the increase in the shear rate. In other words, the samples all exhibited non-Newtonian fluid properties (which refer to the fluid with a nonlinear correlation between the shear rate and shear stress) and shear-thinning behaviour, which presented the pseudo-plastic behaviour [42]. An increasing shear stress was observed with an increasing shear rate among the LLB samples under different homogenisation pressures. Specifically, the apparent viscosity of the samples under 20~100 MPa was consistently higher than that of the sample under 0 MPa throughout the shear rates we observed. The shear stress increased significantly (*p* < 0.05) in the range of 0~60 MPa, which might be due to the interaction force between particles and the formation of interconnected gel networks, thereby enhancing the internal structure of the samples [43]. However, the shear stress did not change significantly (*p* > 0.05) with an increase in pressure of 60~100 MPa because changes in the shear stress might be related to the differences in particle volume fractions, power of interparticle interactions, and particle sizes, just as in the previous results of particle sizes. There was no significant difference (*p* > 0.05) in the homogenisation pressure of 60~100 MPa, which was consistent with the results of studies on walnut yogurt and pomelo peel flour [44,45].

Unlike the changes in shear stress, the apparent viscosity of all homogenously treated samples decreased with an increasing shear rate, exhibiting shear thinning. The difference in apparent viscosity among the samples under different homogeneous pressures at high shear rates was less than that of the samples at low shear rates because the gel structure tended to break at higher shear rates [46], which could also be explained by the structural damage and rearrangement caused by shear rates [33]. The increase in the apparent viscosity of LLBs treated through homogenisation was partly due to a reduction in the suspended particles in the system, similar to the observations of the microstructures and PSD. The decrease in particle size produced a larger interfacial area as well as a decrease in the average distance among particles, leading to an increase in interparticle interactions with increasing homogeneous pressure and a better dispersion of smaller particles in the beverage system, which resulted in higher viscosity values [24]. This result is consistent with the viscosity variation trend of tomato juice and soybean yogurt treated using different homogenisation methods [31,47].

#### 3.3.2. Dynamic Shear Properties

In addition, the deformation resistance of the LLB samples was determined using a dynamic frequency sweep assay. G′ and G″ were used to represent the storage modulus (elastic capacity) and loss modulus (viscous capacity), respectively [34]. G′ and G″ had an increasing scanning frequency (in the range of 0.1~100 rad·s^−^^1^), and G′ was always higher than G″, indicating that all samples had significant viscoelasticity (*p* < 0.05) and that homogenisation had a greater effect on elasticity than the viscosity of LLBs (Figure 5). The LLB can be described as a so-called weak gelatinous structure [48].

In this study, G′ and G″ significantly increased (*p* < 0.05) in the range of 0~60 MPa, indicating that the elasticity and viscosity of LLBs were significantly improved through homogenisation within this homogenisation pressure range. However, with a further increase in pressure, the increases in G′ and G″ were insignificant (*p* > 0.05) within 60~100 MPa, indicating that the continued increase in pressure had less effect on the viscoelasticity of LLBs. The increase in G′ and G″ might be due to starch gelatinisation and protein denaturation in the beverage system caused by homogenisation pressure [43]. It has been previously reported that differences in flow characteristics might be due to the solubilisation of large particles such as starch and pectin in the system or the changes in beverage particle morphology and the interactions among the particles affected by homogenisation [48]. Tan et al. explained that the differences in the viscoelasticity of tomato juice caused by different homogenisation treatments resulted from the decomposition of the system-suspended particles during processing [49]. Therefore, our results showed that homogenisation had an ameliorating effect on the viscoelasticity of LLBs and was most effective at 60 MPa, which meant that the deformation resistance of the beverage was related to different pressures. Hu et al. showed that the G′ and G″ values of mango juice increased with the gradual increase in the range of test frequencies (0.1–10 rad·s^−^^1^); the G′ was always greater than G″ among all samples, and mango juice had a weak gelling property [50]. Bi et al. showed that an increase in homogenisation pressure significantly enhanced the viscoelasticity of soybean-isolated protein emulsion gels and that a certain pressure (less than 60 MPa) contributed to the formation of a more stable three-dimensional network structure of soybean-isolated protein emulsion gels [51]. However, when the homogenisation pressure was too high (up to 80 MPa), the stability of the emulsion structure was affected. Luo et al. found that the homogenised quinoa protein samples had weak gel properties and that G′ and G also increased gradually with increasing pressure (0~50 MPa) [16].

### 3.4. Pulp Sedimentation

Pulp sedimentation is a common problem during the production of LLBs. The sedimentation test results after homogenisation treatment are shown in Figure 6 and Figure 7, where the sample under 0 MPa shows accelerated sedimentation compared with those under 20~40 MPa. The highest sedimentation rate of the sample under 0 MPa was 18%, measured after 0 h of storage (Figure 6a). A clear stratification between the supernatant and turbidity could be observed after 48 h of storage (Figure 7), in which the sample supernatant under 0 MPa was clear, and the most sediment was 21% (Figure 6b). While the supernatant of the samples under 20~100 MPa was translucent, the turbidity or sediment stratification was not apparent, the overall turbidity was higher, and the sedimentation rate was lower compared with the sample under 0 MPa. It might be due to the fact that pectin, cellulose, hemicellulose, proteins, and other components of fruits and vegetables were broken down into small particles through homogenisation, forming smaller particles and tending to be more suspended as the rotational speed increased [52]. The IS of each treatment tended to stabilise after 96 h, with the highest sedimentation rate being 25% for the sample under 0 MPa and the least sedimentation being 9% for those under 40~80 MPa. In contrast, the sedimentation rate of the samples gradually increased when the pressure exceeded 60 MPa (Figure 6c). The same result was verified in the sedimentation rate test, with significant differences (*p* < 0.05) between samples under 20 to 100 MPa and the sample under 0 MPa (Figure 6d). Similar results were found by Silva et al. in the homogenisation effect on the stability of pineapple pulps, in which only the unhomogenised samples showed phase separation during the first 24 h of the sedimentation test [9].

On the other hand, we observed that the sample under 0 MPa had the highest sedimentation rate within 96 h, with the sediment increasing from 18% to 25%, while the high-pressure-treated samples had a lower sedimentation rate, with an increase of only 5% in the precipitation rate of the samples under 60 MPa, suggesting that the samples treated by homogenisation had better stability. According to Stokes’ Law, the sedimentation rate of ions is proportional to the diameter of the particles and the density difference between particle density and fluid, which is inversely proportional to the fluid viscosity [23]. Therefore, the changes in IS during homogenisation are related to the particle size and stability of the homogenised samples. These results are consistent with the sedimentation law of cashew apple juice in a study by Leite et al. [53].

### 3.5. Turbidity

Turbidity reflects the stability of a beverage, which means that the mouthfeel and appearance of beverages with high turbidity are relatively good and more easily accepted by consumers [49]. The turbidity of the sample under 0 MPa was the lowest (1.06%), and that of the sample under 60 MPa was the highest (9.08%). In addition, it showed an upward trend when the homogenisation pressure was between 20 and 60 MPa, while it gradually decreased when the pressure exceeded 60 MPa (Table 2), probably because smaller particles were lighter to pass through the samples as the pressure increased [54]. Studies have shown that homogenisation treatment has an impact on beverage turbidity; the turbidity change in beverages is mainly due to the fact that during the homogenisation process, the particulate matter in the beverages composed of pectin, fat, cellulose compounds, proteins, and their complexes with various substances are broken down into micron sizes to produce a stable dispersion, thus giving the juices higher turbidity [55]. The results of our study are similar to those of Tian et al., indicating that the turbidity of juices can increase after homogenisation treatment and thus improve stability [56].

### 3.6. Colour Value

Colour is one of the important factors affecting consumers’ acceptance of food, which reflects the maturity and freshness of vegetable beverages as an indicator of food safety [57]. Table 3 shows the variations in the parameters L* (lightness), a* (redness: green to red), b* (yellowness: yellow to blue), and ΔE (total colour change) under homogenisation pressure. With an increase in pressure, the value of L* showed a continuous upward trend within the range of 0~60 MPa, with the brightness reaching a maximum of 37.23 at 60 MPa. However, when the pressure increased to 80 MPa, the L* parameter realised a slight decrease, but the brightness was still significantly higher than that of the sample under 0 MPa (*p* < 0.05). This result is consistent with that of Vasquez-Rojas’ study, where it is stated that lily bulbs and lotus seeds are broken into smaller particles through homogenisation treatment, making these particles more light-scattered and reflective [58].

In addition, the brightness of the LLBs gradually decreased when the pressure was higher than 60 MPa, probably due to the increase in temperature resulting from the excessive homogenisation pressure, which led to the formation of the Maillard reaction and browning compounds as well as a colour change in the beverages. The results are similar to those of a study by Xia et al. [59]. On the other hand, all LLB samples had negative values with the parameters a* and b*, indicating that green and blue were the primary sources of the colour parameters of LLBs, and as the homogenisation pressure increased, the a* and b* values were significantly lower compared to those of untreated samples (*p* < 0.05). ΔE is an important parameter for colour difference analysis. Consistent with the parameters L*, a*, and b*, the maximum colour difference of 4.47 was observed at a pressure of 60 MPa, which was significantly higher compared to 0 MPa (*p* < 0.05).

In conclusion, the homogenisation treatment effectively changed the colour value of LLBs and enhanced the brightness of the samples in the low-pressure range. However, excessive pressure may cause mechanical damage and destroy the cell structure, and due to the increase in pressure, the particles within beverages are broken into smaller diameters, resulting in excessive contact of the particles with oxygen. In contrast, the increase in pressure leads to an increase in the temperature of the beverages, producing non-enzymatic browning, which is not conducive to the enhancement of brightness and ultimately affects the change in colour value.

### 3.7. pH and TSS

The pH value is an important parameter for assessing the quality of beverages, which directly affects the storage conditions and preservation methods. The results showed that the pressures of all samples treated through homogenisation were significantly lower than 0 MPa (*p* < 0.05), and the pH value was the lowest at 100 MPa, which was 0.13 lower than that of the sample under 0 MPa (Table 2). Due to the pressure generated during homogenisation, the particle size of the beverages is reduced, resulting in an increase in the exposed surface area of the particles and a change in pH, which is influenced by charged molecules. On the other hand, homogenisation pressures and the accompanying high temperatures may change the conformation of proteins, affecting the charge or solubility and leading to a change in the pH value of the beverages [60]. Gul and Saricaoglu et al. studied the pH of hazelnut beverages and found that the pH was reduced after homogenisation treatment, which they attributed to changes in the dissociation constants of acids and bases to the applied homogenisation pressure [33]. Homogenisation significantly affected the TSS of LLBs (*p* < 0.05), and the TSS content gradually increased with homogenisation pressure. The highest TSS content was 6.33 ◦Brix at 100 MPa, which was 1.35 ◦Brix higher than that at 0 MPa. Similar results were obtained in a previous study on hazelnut milk, and the increase in sugar content might be due to the release of starch granules under mechanical pressure or an increase in the proportion of soluble protein in the homogenisation treatment [46]. The effect of the homogenisation pressures of 0, 20, 40, 60, 80, and 100 MPa on lily pulps was investigated by Liu et al. [30]. Lily pulps were reported to be a composite suspension system consisting of starch-filled cells, parenchymal cells, or aggregates of cells dispersed in liquids. Stress and high-temperature cracking of these cells during homogenisation resulted in an increased starch grain expansion and gelatinisation degree, with a corresponding increase in the TSS content.

### 3.8. Ascorbic Acid Content

The stability of ascorbic acid is susceptible to degradation by high temperature, light, and oxygen. Table 2 shows the AAC of LLBs. The sample under 0 MPa (13.88 mg·100 mL^−^^1^) is significantly higher than the treatment groups under 100 MPa (11.05 mg·100 mL^−^^1^); however, when the homogenisation pressure is in the range of 20~80 MPa, the AAC is only slightly reduced, which shows no significant difference from the sample under 0 MPa (*p* > 0.05). Similar results were reported by Pérez-Conesa et al., who treated tomato puree using a homogeneous pressure of 10~20 MPa, and the AAC did not reduce obviously, probably due to the low pressure used for the homogenisation treatment [61]. Saricaoglu et al. reported that the AAC of rosehip nectar was significantly reduced when homogenised thrice under 75 MPa or once under 155 MPa [24]. Velázquez-Estrada et al. found that the AAC of orange juice was reduced under a homogenisation pressure of 100~200 MPa and that the degree of oxidation of ascorbic acid was directly proportional to the homogenisation pressure [62]. It is well known that the high temperature generated by homogenisation pressure accelerates the oxidation of ascorbic acid [40]. However, the temperature might not be the only factor affecting the reduction of ascorbic acid in this experiment, as relevant studies have shown that heavy metal cations in the homogeniser also accelerate its oxidative degradation [63].

### 3.9. Sensory Evaluation

Sensory evaluation is a qualitative and quantitative measurement as well as analysis of food products with the help of people’s five senses, psychology, and physiology, which helps to ensure consumer acceptance and satisfaction with product quality. The related attributes of appearance, colour, aroma, taste, mouthfeel, and overall acceptability were included in the sensory evaluation of the LLB (Figure 8). Appearance and colour are important aspects in evaluating beverages. Samples homogenised at 20~100 MPa were all significantly different (*p* < 0.05) compared to the sample under 0 MPa, which was consistent with the results of turbidity and IS, indicating that the sample under 0 MPa simply flowed through the homogeniser without producing changes in its stability. Aroma is an important part of sensory evaluation—the changes that will affect the acceptance of consumers. Our results showed no significant difference among the aromas of the homogenised samples (*p* > 0.05), probably due to the fact that the two vegetables, lotus seeds and lily bulbs, had less pronounced aromas of their own. Mouthfeel and taste scores indicate whether a beverage has a balanced sugar/acid ratio and a pleasant fruit or vegetable flavour. The homogenised samples had a significantly better (*p* < 0.05) mouthfeel compared to the sample under 0 MPa, indicating that the homogenisation treatment had a positive effect on the mouthfeel of LLBs. However, homogenisation had an insignificant effect on taste, and there was an insignificant difference between samples under 20 to 100 MPa compared with the sample under 0 MPa (*p* > 0.05). Regarding the overall receptivity, the judges were satisfied with the LLB samples, with the highest receptivity in the samples under 60 and 80 MPa.

## 4. Conclusions

In this study, we compared a LLB treated under different homogenisation pressures (0~100 MPa) to assess the effect of homogenisation on its physical properties and sensory quality. Several assays were performed, including PSD, microstructure, rheological behaviour, IS, turbidity, and sensory evaluation. In conclusion, the homogenisation treatment caused the suspended particles in the beverages to be strongly decomposed under high pressure, increasing the viscosity of the samples and causing the beverages to appear to have weak gel properties. At the same time, homogenisation improved the turbidity and stability of the LLB, which had a positive effect on IS. The overall acceptance of the LLB sample under 60 MPa by the judges was the highest in the sensory evaluation. Homogenisation did not disturb the original PH, TSS, or AAC in LLBs. Thus, homogenisation is a quite suitable pretreatment technology for enhancing the physical properties and sensory quality of LLBs. It should be noted that there were some limitations in this study; we only investigated the effect of different pressures on the physical properties of LLBs. Further investigations of the enzyme activities, bioactive substances, and shelf life under different homogeneous conditions are recommended.

## Figures and Tables

**Figure 1 foods-13-00769-f001:**
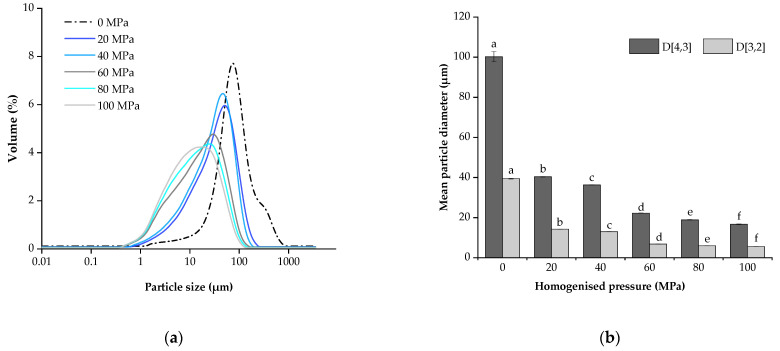
The particle size of lotus seed and lily bulb beverage: (**a**) particle size distribution (PSD); (**b**) volume-based diameter D [4,3], and area-based diameter D [3,2]. Vertical bars indicate ± standard errors of the mean values (n = 3), the same as below. Different letters in the same row indicate significant differences (*p* < 0.05) between the means.

**Figure 2 foods-13-00769-f002:**
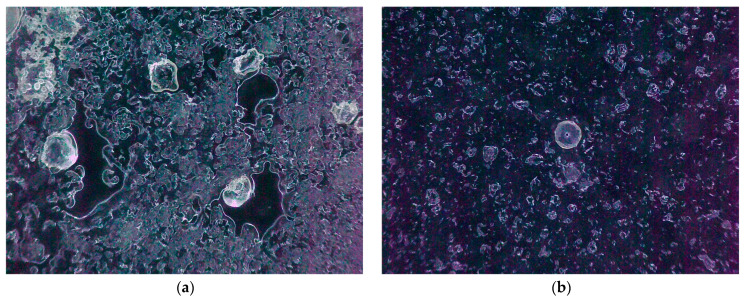
Plate of electron microscope test of lotus seed and lily bulb beverage by homogenisation: (**a**) 0 MPa; (**b**) 20 MPa; (**c**) 40 MPa; (**d**) 60 MPa; (**e**) 80 MPa; and (**f**) 100 MPa.

**Figure 3 foods-13-00769-f003:**
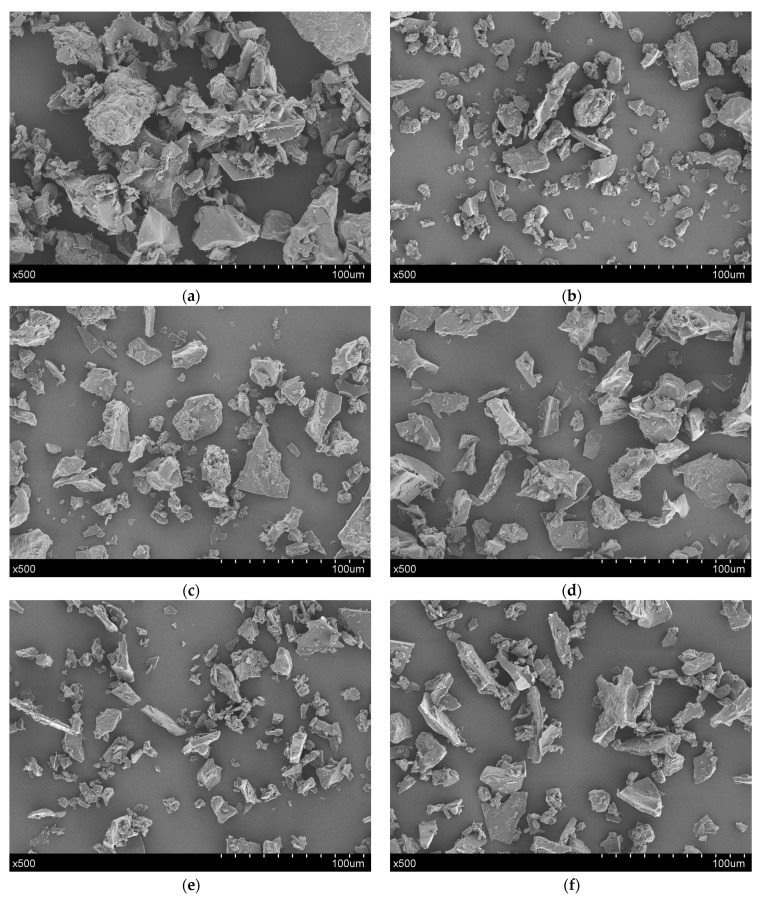
Plate of scanning electron microscope test of lotus seed and lily bulb beverage by homogenisation: (**a**) 0 MPa; (**b**) 20 MPa; (**c**) 40 MPa; (**d**) 60 MPa; (**e**) 80 MPa; and (**f**) 100 MPa.

**Figure 4 foods-13-00769-f004:**
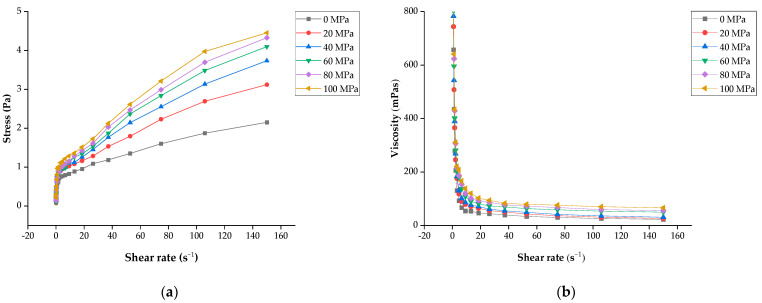
Effect of different shear rate conditions of lotus seed and lily bulb beverage: (**a**) the relationship between shear rate and shear stress; (**b**) the relationship between shear rate and apparent viscosity. Different letters in the same row indicate significant differences (*p* < 0.05) between the means.

**Figure 5 foods-13-00769-f005:**
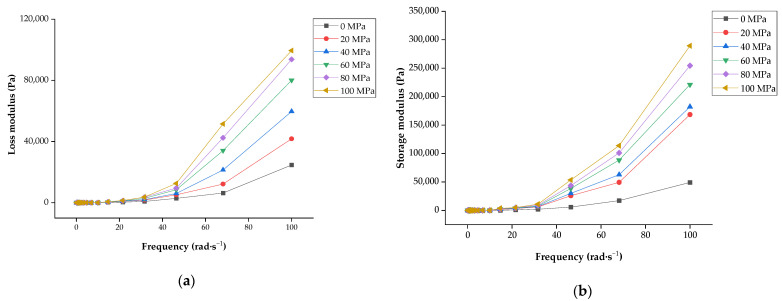
Effect of different scanning frequencies on the viscoelasticity of lotus seed and lily bulb beverage: (**a**) the relationship between scanning frequency and loss modulus (G″); (**b**) the relationship between scanning frequency and storage modulus (G′). Different letters in the same row indicate significant differences (*p* < 0.05) between the means.

**Figure 6 foods-13-00769-f006:**
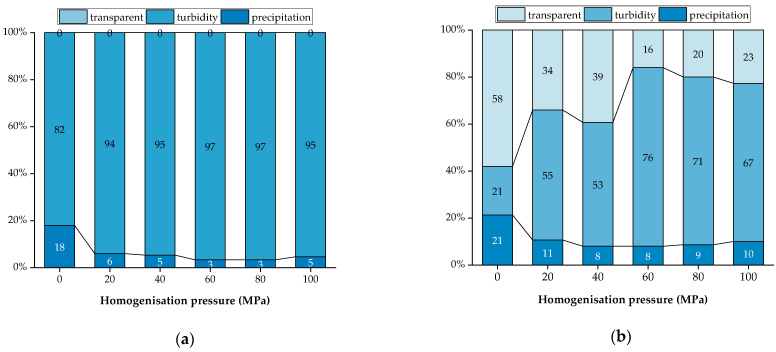
Proportion (%) of turbidity, transparency, and precipitation in lotus seed and lily bulb beverage samples: (**a**) 0 h; (**b**) 48 h; (**c**) 96 h; and (**d**) sedimentation index. Different letters in the same row indicate significant differences (*p* < 0.05) between the means.

**Figure 7 foods-13-00769-f007:**
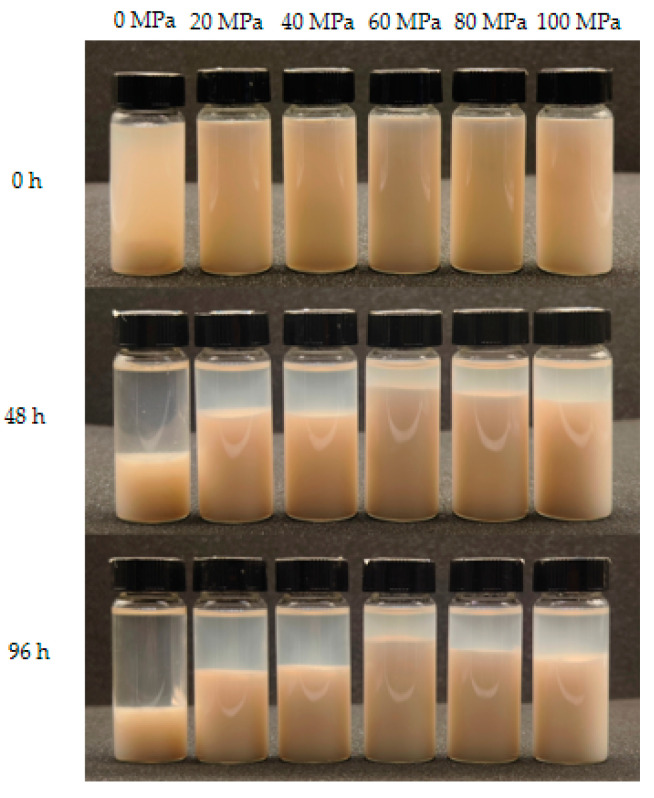
Lotus seed and lily bulb beverage samples (different homogenisation pressures: 0 MPa, 20 MPa, 40 MPa, 60 MPa, 80 MPa, and 100 MPa) were stored at 0 h, 48 h, and 96 h.

**Figure 8 foods-13-00769-f008:**
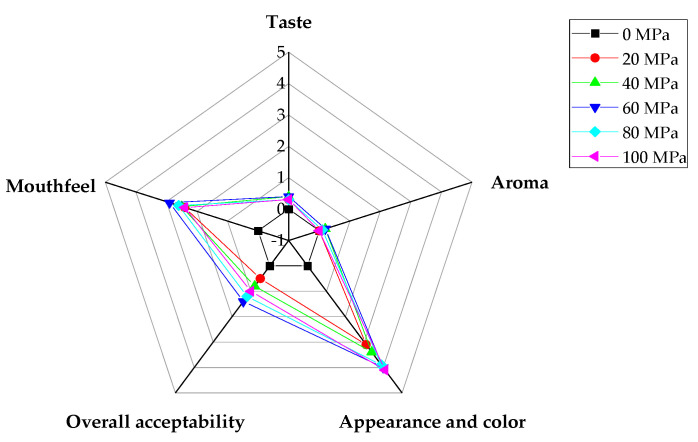
Sensory parameters of lotus seed and lily bulb beverage.

**Table 1 foods-13-00769-t001:** Particle size characteristics in lotus seed and lily bulb beverage. Different letters in the same row indicate significant differences (*p* < 0.05) between the means.

Homogenisation Pressure	D10 (μm)	D50 (μm)	D90 (μm)	Particle Distribution Span
0 MPa	25.41 ± 0.02 ^a^	71.06 ± 0.29 ^a^	218.43 ± 9.99 ^a^	2.72 ± 0.13 ^e^
20 MPa	6.44 ± 0.02 ^b^	32.92 ± 0.15 ^b^	84.69 ± 0.32 ^b^	2.38 ± 0.01 ^d^
40 MPa	5.83 ± 0.02 ^c^	31.22 ± 0.13 ^c^	73.84 ± 0.15 ^c^	2.18 ± 0.05 ^c^
60 MPa	2.68 ± 0.02 ^d^	15.82 ± 0.13 ^d^	51.27 ± 0.16 ^d^	3.07 ± 0.02 ^b^
80 MPa	2.43 ± 0.02 ^e^	12.67 ± 0.07 ^e^	44.55 ± 0.34 ^de^	3.32 ± 0.01 ^a^
100 MPa	2.33 ± 0.02 ^f^	10.94 ± 0.04 ^f^	39.49 ± 0.50 ^e^	3.40 ± 0.03 ^a^

**Table 2 foods-13-00769-t002:** Other physicochemical characteristics and physical properties of lotus seed lily bulb beverage. Different letters in the same row indicate significant differences (*p* < 0.05) between the means.

Homogenisation Pressure	Relative Turbidity (%)	PH	TSS (◦Brix)	Ascorbic Acid (mg·100 mL^−1^)
0 MPa	1.06 ± 0.16 ^c^	4.61 ± 0.03 ^a^	4.98 ± 0.31 ^c^	13.88 ± 1.78 ^a^
20 MPa	5.15 ± 0.11 ^b^	4.55 ± 0.05 ^b^	5.63 ± 0.31 ^b^	13.60 ± 1.76 ^a^
40 MPa	5.29 ± 0.20 ^b^	4.54 ± 0.04 ^b^	5.73 ± 0.25 ^b^	13.32 ± 1.28 ^a^
60 MPa	9.08 ± 1.09 ^a^	4.51 ± 0.04 ^b^	6.27 ± 0.15 ^a^	13.03 ± 1.08 ^a^
80 MPa	8.88 ± 0.76 ^a^	4.49 ± 0.05 ^b^	6.30 ± 0.26 ^a^	12.47 ± 0.88 ^ab^
100 MPa	8.73 ± 0.29 ^a^	4.48 ± 0.03 ^b^	6.33 ± 0.25 ^a^	11.05 ± 0.69 ^b^

**Table 3 foods-13-00769-t003:** L*, a*, b*, and ΔE of lotus seed lily bulb beverage. Different letters in the same row indicate significant differences (*p* < 0.05) between the means.

Homogenisation Pressure	L*	a*	b*	ΔE
0 MPa	32.73 ± 0.54 ^d^	−0.58 ± 0.35 ^a^	−3.79 ± 0.02 ^a^	0.00 ± 0.00 ^d^
20 MPa	35.88 ± 0.02 ^b^	−0.77 ± 0.02 ^bc^	−4.34 ± 0.08 ^b^	3.20 ± 0.03 ^b^
40 MPa	36.31 ± 0.45 ^b^	−0.73 ± 0.02 ^b^	−4.30 ± 0.02 ^b^	3.62 ± 0.45 ^b^
60 MPa	37.23 ± 0.51 ^a^	−0.83 ± 0.02 ^e^	−4.59 ± 0.05 ^c^	4.47 ± 0.50 ^a^
80 MPa	36.44 ± 0.04 ^ab^	−0.79 ± 0.02 ^de^	−4.51 ± 0.04 ^c^	3.79 ± 0.05 ^b^
100 MPa	35.09 ± 0.41 ^c^	−0.73 ± 0.02 ^bc^	−4.34 ± 0.10 ^b^	2.42 ± 0.41 ^c^

## Data Availability

The original contributions presented in the study are included in the article, further inquiries can be directed to the corresponding author.

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
