# Peer review of "Physical Stability of Lotus Seed and Lily Bulb Beverage: The Effects of Homogenisation on Particle Size Distribution, Microstructure, Rheological Behaviour, and Sensory Properties"

_foods, 2024, doi:10.3390/foods13050769_

Round 1

Reviewer 1 Report

Comments and Suggestions for Authors

The manuscript entitled: " Physical Stability of Lotus Seed and Lily Beverages (LLB): Effects of High-Pressure Homogenization (HPH) on Particle Size Distribution, Microstructure, Rheological Behaviour, and Sensory Properties" submitted to Foods.

 Main comment:

The authors carried out experiments using pressure: 0-100 MPa. In this case, the phrase HPH cannot be used because a pressure of 100–200 MPa is generally used in HPH, and 200-400 when we talk about UHPH. Please remove some of the descriptions regarding HPH and this nomenclature.  

Abstract:

·         pH changes were not statistically significant between samples at different pressure levels, so I don't think this is important news in the abstract

·         In my opinion, the first and the last sentence is unnecessary

Introduction:

·         To determine whether a given raw material is rich in a certain ingredient, you should provide its content. In lines 39-40 you write that "Lilium brownii var. viridulum (...) is rich in proteins, vitamins, amino acids, and various bioactive components", but you do not provide the values. Perhaps these are very minor contents

·         Lines 66-69 this is very brief, because e.g. extending the shelf-life is a consequence of the reduction of microorganisms in HPH

·         Lines 75-77: “Wellala et al. found that HPH improved rheological properties, cloud stability, and microbial populations in blended juices consisting of apple, peach, and carrot 17.” – my question is: how hph improved microbial populations in this case?

·         Lines 77-79: what was the result of Liu's research?

·         Lines 80-81: I do not agree that in the case of beverages the impact of HPH on their quality is less known.

Methodology:

·         Section 2.3: Which sample is the control sample: fresh or pasteurized?

·         Line 110: On what basis was the homogenization time determined to be 5 min? Does the liquid flow through the homogenizer or is it placed in some glass/vessel?

·         Lines 113-115: The rule is to write deviations with the same number of digits after the decimal point as the main number, e.g. 14,65 ± 0,21 (not 20.5 ± 0.50).

·         Lines 115-117: Are the results in this article presented for samples after pressure treatment and then pasteurization? This is a methodological error if we want to talk about what changes were caused by the use of pressure, while pasteurization can lead, for example, to aggregation of particles.

·         “2.4. Particle Size Determination” should be “2.4. Particle Size Distribution”

·         In my opinion equation 1 and 2 are unnecessary

·         “The steady-state rheological test was performed on LLB in rotating mode, temperature (25°C), shear rate (0.1-150 s-1), shear rate and shear stress, shear rate, and apparent viscosity flow curves were drawn.” – this sentence is unnecessary, because your results are in another section

·         Line 164: “by the following equations” should be “by the following equation”

·         Section 2.8: What is the unit of turbidity in this study?

·         Section 2.9: Is the analysis of color parameters not performed in accordance with the device's instructions?

·         Lines 180-181: You forgot to mention that the samples were also pasteurized.

·         Section 2.10: reference to Vieira et al. 22 is unnecessary - pH is the simplest analysis performed according to the device instructions

·         Section 2.11: The results are expressed in what units?

·         Section 2.12: Who were the evaluators, were they students or a qualified trained team?

·         Lines 205-206: “The result was based on an intensity scale from 4 to 4 points” – it is correct?

·         Section 2.12: You mentioned that samples were compared to the unhomogenized sample, but what about pasteurization influence on sensory quality?

Results:

·         Section 3.1: Particle distribution span is not described in text

·         Lines 358-360: Please expand on other authors' results

·         Lines 378-379: Which conditions was used by Silva et al. and which results he got?

·         Line 467: Please reword this sentence

·         since you describe that after applying pressure, the color is less acceptable to consumers, I see a lack of consistency here, because in section 3.9 and figure 10 it turns out that these samples are rated better. what is the reason for this?

Conclusions:

·         Conclusions should not be a abstract, so what conclusions can be drawn from this research?

 References:

·         Despite a good literature review, I think that there should also be authors who were among the first to write about HPH e.g. Suárez-Jacobo team, Calligaris team, Donsì but also newer authors with good quality publications, e.g. SzczepaÅ„ska, Patrignani

 Others:

·         References in the text should be in square brackets

·         “Mpa” should be “MPa”

Author Response

Dear reviewer,

As per your kind guidance. We have carefully revised our manuscript according to these suggestions. ‘Point-by-point response letter for reviewer’ have been uploaded to the attachment in word file. Please see the attachment.

We hope the revised manuscript now meets the standard for publishing in Foods.

Thanks for your consideration.

Jiajia Su

Reviewer 2 Report

Comments and Suggestions for Authors

The objective of this study is to investigate the impact of high-pressure homogenization (HPH) on lotus seed and lily beverage (LLB). The research was well-designed and discussed.

However, the authors justify changes in color, pH, total soluble solids, and ascorbic acid content based on the temperature levels utilized in HPH. In this regard, did the authors conduct any experiments to discern the individual contributions of HPH and temperature to the properties of the beverages?

Minor comments:

Page 5, line 206: correct “scale from 4 to 4 points”

Page 12, line 363: correct "Figures 6 and 7”

Page 15, line 471: correct " Table 2”

Page 16, line 503: the authors refer that the scores ranging from 7.7 to 8.1 but the intensity scale range from  -4 to 4 points.

Comments on the Quality of English Language

 Minor editing of English language required

Author Response

Dear reviewer,

As per your kind guidance. We have carefully revised our manuscript according to these suggestions. ‘Point-by-point response letter for reviewer’ and have been uploaded to the attachment in word file. Please see the attachment.

We hope the revised manuscript now meets the standard for publishing in Foods.

Thanks for your consideration.

Jiajia Su

Round 2

Reviewer 1 Report

Comments and Suggestions for Authors

In my opinion, the manuscript can be published in its current version. I have no comments or additional comments.